# Lentiviral Vectors for Delivery of Gene-Editing Systems Based on CRISPR/Cas: Current State and Perspectives

**DOI:** 10.3390/v13071288

**Published:** 2021-07-01

**Authors:** Wendy Dong, Boris Kantor

**Affiliations:** 1Department of Neurobiology, Duke University Medical Center, Duke University, DUMC Box 3209, Durham, NC 27710, USA; wendy.dong@duke.edu; 2Viral Vector Core, Duke University Medical Center, Durham, NC 27710, USA; 3Duke Center for Advanced Genomic Technologies, Durham, NC 27710, USA

**Keywords:** lentiviral vectors, intergrase-deficient lentiviral vectors, CRISPR/Cas9 systems, base-editing, prime-editing, genome-editing, epigenome-editing, vector mediated gene-to-cell transfer, gene therapy

## Abstract

CRISPR/Cas technology has revolutionized the fields of the genome- and epigenome-editing by supplying unparalleled control over genomic sequences and expression. Lentiviral vector (LV) systems are one of the main delivery vehicles for the CRISPR/Cas systems due to (*i*) its ability to carry bulky and complex transgenes and (*ii*) sustain robust and long-term expression in a broad range of dividing and non-dividing cells in vitro and in vivo. It is thus reasonable that substantial effort has been allocated towards the development of the improved and optimized LV systems for effective and accurate gene-to-cell transfer of CRISPR/Cas tools. The main effort on that end has been put towards the improvement and optimization of the vector’s expression, development of integrase-deficient lentiviral vector (IDLV), aiming to minimize the risk of oncogenicity, toxicity, and pathogenicity, and enhancing manufacturing protocols for clinical applications required large-scale production. In this review, we will devote attention to (*i*) the basic biology of lentiviruses, and (*ii*) recent advances in the development of safer and more efficient CRISPR/Cas vector systems towards their use in preclinical and clinical applications. In addition, we will discuss in detail the recent progress in the repurposing of CRISPR/Cas systems related to base-editing and prime-editing applications.

## 1. HIV-1 Based Vectors: Basic Biology

As a type of simple retrovirus, HIV-1 derived lentiviruses are capable to hijack host-mediated machinery to sustain an efficient nuclear import across the intact nuclear membrane [1]. This feature has allowed them to efficiently transduce nondividing and terminally differentiated cells (e.g., postmitotic neurons, hepatocytes, or macrophages) with superb efficiency (reviewed in [2]). The long-lasting effect of the viral transduction supports long-term production of the therapeutic gene-of-interest (thus providing permanent steady-state “dosing” after a single administration of the virus) which is essential for gene therapy applications. LV genome occupies ~10.7 kb of single-stranded RNA (ssRNA) placed inside a lipid-enriched spheric capsid of ~100 nm in diameter (Figure 1). The viral genome encodes both structural and enzymatic genes including *gag* and *pol*. The polycistronic *gag gene* encodes three products, namely matrix (MA), capsid (CA), and nucleoproteins (NC). The polycistronic *pol gene* supplies three viral enzymes, namely reverse transcriptase (RT), protease (PR), and integrase (IN) (Figure 1A). LV is an enveloped virus that uses a glycoprotein envelope to attach and enter the host cell. The creation of heterologous envelopes used for viral particle pseudotyping was one of the main progresses in the field that allowed to dramatically diversify and extend the tropism of transduction. Additionally, the supplementation of viral particles with heterologous envelopes has positively impacted vector safety (reviewed in [2]). As mentioned above, LVs are capable of being efficiently pseudotyped with a broad range of heterologous envelopes, enabling broad viral tropism. For example, LV supplemented with Mokola virus (MV), Ross River virus (RRV), and Rabies virus (RV) demonstrated a strong preference for the transduction into neuronal cells (reviewed in [3]). However, the most common envelope used to pseudotype viral particles is that of vesicular stomatitis virus protein G (VSV-G). The envelope has been shown to support an extremely broad range of tropism and as such is used for transduction into most cells and tissues (reviewed in [2]). In addition to *gag* and *pol,* lentiviruses carry six supplementary genes: *rev* and *tat,* involved in viral transcription and export, respectively, and *nef*, *vif*, *vpr*, and *vpu*, involved in viral entry, assembly, replication, particle formation and release (Figure 1) and [4]. It is important to note that the latter four accessory products are dispensable for vector production, and as such can be omitted from the packaging cassette of the vector (Figure 2A). The removal of these products has been shown to have a positive effect on the vector safety; importantly, their deletion also creates space for cloning larger inserts [5,6,7,8]. Indeed, the second generation of the packaging system carried only the *tat* and *rev* genes (Figure 2B) and [9]. The third generation of the packaging system does not have the *tat* gene, which is in co-incidental with the deletion of the endogenous promoter harboring *tat-*responsive element, TAR, at the U3′ region of the LTRs (Figure 2C). Instead, full-length RNA of the virus is transcribed from the strong ubiquitous promoter derived from Rous sarcoma virus (RSV) or cytomegalovirus (CMV). Further improvement of virus safety has been achieved with the development of the fourth-generation packaging plasmid, highlighted by the split of the *gag/pol* and *rev* sequences into two different cassettes (Figure 2D). In fact, the fourth generation of the packaging systems is the safest to date [7]. It should be noted that *rev* gene is present in all the packaging systems, as its product REV plays a key role in exporting fully length and partially spliced viral RNA (vRNA) from the nucleus into the cytoplasm [10]. The advanced generations of the packaging plasmids also harbor a strong, heterologous poly-adenylation signal (poly-A), derived from the SV40 virus or bovine/human growth hormone (bGH/hGH). These potent poly-As enable high level of vRNA stability, and as such their inclusion has been found to be advantageous for the packaging and viral titers [7,10]. Moreover, the inclusion of the woodchuck hepatitis virus posttranscriptional regulatory element (WPRE) and the central polypurine tract (cPPT) into the viral transfer cassette has been shown to further improve vRNA stability, transcription efficiency, and overall viral titer [11,12]. Importantly, the above modifications greatly reduce the likelihood of the appearance of recombination-competent retroviruses (RCR), which positively impacts viral safety characteristics.

## 2. The Life Cycle of Lentiviral Vectors

Attachment, binding and entry of the lentivirus is mediated via interaction between the envelope protein and its receptor. Following entry and uncoating, the reverse transcriptase enzyme (RT) is set to start the reverse transcription reaction. The product of the RT reaction is double-stranded (ds) linear DNA. The first step in the RT process depends on two cis-acting elements located within the viral RNA: the primer binding site (PBS) and the polypurine tract (PPT). The PBS is recognized by tRNA^Lys3^_,_ which supplies a specific primer for the RT-mediated amplification. The PPT element comprises a purine-rich stretch of nucleotides that are resistant to RNase H-mediated degradation of positive strand of vRNA; the undegraded PPT sequence can then act as a primer towards the positive strand synthesis. As mentioned above, the completion of the RT reaction results in double stranded, linear DNA (reviewed in [2]). It is important to note that the vDNA has the same sequence as the full-length vRNA; the only difference between them is that vDNA carries two U3′, and two U5′ regions placed on both sides of the LTRs; whereas vRNA harbors a single U3′ and U5′ region. The U3′ region on the 5′-LTR carries the viral promoter (RSV or CMV), whereas U5′ of the 3′-LTR harbors the poly-A signal (reviewed in [2]). After the RT reaction is completed, the dsDNA undergoes nuclear import; this process is mediated by host-derived importin complexes. Following nuclear transportation, the vDNA serves as a precursor for viral integration. The *int* gene encodes the protein integrase (IN), which mediates the integration process by binding to and cleavage within the *att* sites located on the LTR ends [13,14,15]. Then, vDNA becomes an integral part of the host’s chromosome; it may replicate alongside the host genome, and can be passed on to the cell’s progeny (Figure 3) [16]. The RT enzyme is essential for vector production, whereas a functional lentivirus can be generated without active IN enzyme. We and others have taken an advantage of the fact that IN is not required for viral production and expression to construct an integrase-deficient lentiviral vector (IDLV), which demonstrates some important advantages over the parental integrase-competent lentiviral vectors (ICLV), discussed below.

## 3. Safety of Lentiviral Vectors

As mentioned above, the use of heterologous promoters, e.g., CMV or RSV, instead of the endogenous HIV-1 promoter located in the 5′-LTR, enabled an independence from the Tat protein, which greatly improved the vector’s safety [17,18,19]. Subsequently and importantly, deletions within the U3′-region of the 3-LTR that harbors the enhancer/regulatory elements’ and the TATA box, helped to develop a self-inactivating (SIN) lentiviral vector [20]. During RT reaction, the U3′ region relocates to the 5′-LTR, as such SIN-vectors are completely devoid of HIV-1 enhancer/promoter sequences. Consequently, SIN-vectors lack the ability of producing a full-length vRNA that could be packaged into the viral particles. It would be important to note that the deletion did not affect vector production, and vector titer remains to be comparable with those of non-SIN counterparts. As mentioned above, development of a SIN vectors further reduced the likelihood of the appearance of RCVs. Furthermore, it has a greatly lower likelihood of mobilizing the vector’s mRNA by the replication-wt virus [21]. In addition, a lack of an enhancer/promoter region diminished the risk of inadvertent activation of silent host promoters by the lentiviral provirus.

## 4. Non-Integrating Lentiviral Vectors

Despite significant improvement in viral safety, simple retroviruses, e.g., γ-retroviruses have a relatively high risk of insertional mutagenesis (reviewed in [2]). In fact, initially successful treatments of ADA-SCID, SCID-X1, and X-linked CGD diseases with γ-retroviral vectors were tragically complicated by blood cancers developed by several patients [22]. It has been reported that the patients carried provirus DNA in the proximity of proto-oncogenes dysregulating their expression [22]. Generally, LV has the same traits, due to the inherent capacity for integration; with that said, the risk of oncogenicity and toxicity of the complex retroviruses seems to be lower than that of γ-retroviruses. In fact, it was reported that γ-retrovirus-transduced hematopoietic stem cells transplanted into tumor-susceptible mice resulted in tumorigenic processes, whereas no adverse effect was shown in the same setting with lentiviruses [23]. On the same note, it has been reported that a substantially larger titer of lentivirus is necessary to reach a similar oncogenic risk to that of γ-retroviral vectors [24]. However, lentiviruses do not completely lack tumorigenicity. For example, it has been demonstrated that the lentiviral vectors of horses, equine-infectious anemia virus (EIAV) vectors, can cause multiple tumors in the livers of mice following in utero and neonatal administrations [25]. A causal link between EIAV vectors and tumorigenesis has yet to be established; however, it is important to note that in the same study the use of HIV-1 based vectors were not linked with formation of any detectable tumors [25]. Importantly, both γ-retroviral vectors and lentiviral vectors have been recently developed into efficient platform utilized for curing monogenic inherited disorders caused by an altered development and/or function of the blood system, such as immune deficiencies and red blood cell and platelet disorders (reviewed in [26]). Thanks to the positive efficacy and safety data from hematopoietic stem progenitor cells (HSPCs) gene therapy trials collected during last decade, two advanced therapies based on γ-retroviral vector-engineered HSPCs have been approved for the EU market, and many other clinical trials are also in advanced stages in the US [26]). In 2016, the European Medicines Agency (EMA) approved Strimvelis for the treatment of ADA-SCID. a replication deficient γ-retroviral vector based on Moloney murine leukemia virus (MMLV) encoding the cDNA sequence for human ADA [27]. Next, in 2019, Zynteglo became the first lentiviral vector-based gene therapy product for transfusion-dependent β-thalassemia patients approved by the EMA [28]. Furthermore, lentiviral-HSPC-based medicine, Libmeldy, most recently received EMA positive opinion for marketing authorization to treat metachromatic leukodystrophy (MLD) [26].

The above applications required stable and long-lasting expression of the therapeutic transgenes supported by integrase-competent retro- and lentiviruses. Nevertheless, permanently expressed CRISPR/Cas systems being seen rather as substantial disadvantage, as stable expression of their components may facilitate undesirable off-target effects, hindering their utility for genome- and epigenome-editing applications [29]. Indeed, the rise in promiscuous interactions with off-target genes due to excess guide RNA (gRNA)/Cas9 is well-documented [29]. Furthermore, sustained expression of gRNA/Cas9 in vitro increases the tolerability of mismatches in the guide-matching region and the protospacer adjacent motif (PAM), thereby promoting non-specific double-strand breaks (DSBs) [30]. Along the same lines, the ratio of insertions and deletions (indels) at off-target versus target sites in vivo increases with higher Cas9 and gRNA concentrations [31]. These results suggest that transient delivery systems utilizing a “hit and run” strategy for terminal modification of the DNA loci would be advantageous for high-precision gene editing.

To that end, we and others worked on the development of an integrase-deficient lentiviral vector (IDLV). IDLVs can be generated by creating non-pleiotropic (class II) mutations within the *int* gene [32]. Point mutations in the catalytic center of the IN enzyme, called the triad sequence, are capable of completely abolishing the integration process, without causing any significant aberration in other steps of the lentiviral life cycle (Figure 3). Furthermore, we previously demonstrated that IDLVs could effectively express their genomes in vitro and in vivo. However, the levels of expression have been found to be lower than those of the integrating counterpart [21,33]. Notably, even that low level of expression has been found to be sufficient to correct human hereditary diseases [34,35]. More recently, we demonstrated that the low level of IDLV-mediated expression is due to the repressive chromatin structure formed around its episomal genomes [21]. Importantly, we demonstrated that low IDLV-expression could be augmented by targeting and deleting (in cis- or in trans-) repressive factors, including histone deacetylases (HDACs). In fact, we demonstrated that the HDAC-inhibition led to substantial activation of viral expression from the episomal genomes in vitro and in vivo [34,35,36]. More recently, we demonstrated that reconstitution and placement of some transcription enhancers, including Sp1 and NF-kB binding sites, into the viral expression cassette could substantially improve the packaging efficiency, titer, and expression of IDLV both in vitro and in vivo [37]. Importantly, these perturbations did not compromise the low integration frequency shown by IDLVs [33,36,38]. We determined this rate to be approximately 1/3850 in HeLa cells in vitro, and 1/111 in mouse cerebellar neurons in vivo [33,36]. These integration rates are ~500-fold lower than that of ICLVs [33,36]. Along the same lines, IDLV caused significantly lower levels of InDel formation and other off-target effects, compared to ICLV, when they were used for delivering CRISPR/Cas9 components in vitro and in vivo [37,39]. The non-integrating platform is equally as efficient as the integrating equivalent in delivering and expressing genome- and epigenome-editing tools in dissociated post-mitotic neurons in vitro and in rat brain neurons in vivo [37,39]. As such, IDLV could become an important platform for CRISPR/Cas9 delivery into the CNS and other tissues and organs.

## 5. Adeno-Associate Vectors (AAVs)

This review is devoted to lentiviral vectors; however, it would be remiss not to mention the most frequently used viral platform for gene therapy, adeno-associated vector (AAV) (reviewed in [2]). AAV is an ideal viral system for several reasons: (i) the vector has no known associated pathologies and causes only a mild immune response in humans; (ii) similarly to IDLV vectors, the AAV genome can be persisted long-termly in episomal forms, and thus presents an opportunity for extended transgene expression in non-dividing cells and tissues [2]. (iii) Lastly, the AAV genomic structure is well-characterized, so the consequences of genome-targeted manipulations can adequately be predicted [2]. For the above considerations, over the last three decades a significant effort has been devoted to developing AAV into one of the gold-standard delivery systems for broad range of gene-therapy applications [40]. Notwithstanding these advances, impressive and rapidly diversifying array of AAV-CRISPR/Cas-derived tools predominantly have been used in vitro (reviewed in [41]). Efficient delivery in vivo using AAV vectors is a significantly more challenging task. In fact, the limited packaging capacity of the AAV genomes is the main bottleneck for its use, in an all-in-one configuration, for delivery of bulky and complex CRISPR/Cas transgenes in vivo. One of the approaches to overcome the significant restraints imposed by AAV’s ~4.7 kb functional packaging capacity is to physically split a CRISPR/Cas transgene into two pieces, which are packaged into separate AAV vectors. The resulting AAVs are then co-delivered, and the complete protein is reassembled in situ by a split intein—a pair of domains which “splice themselves out”, thus re-joining two peptides in the end-to-end configuration [42]. Nevertheless, further improvement of this and other systems (reviewed in [41]) would be necessary to achieve the desired therapeutic efficacy.

## 6. Overview of CRISPR/Cas9-Based Gene-Editing Systems

The clustered regularly interspaced short palindromic repeats (CRISPR) and CRISPR-associated protein (Cas) system has recently evolved to be a revolutionary platform for both genome- and epigenome-editing manipulations in a broad range of tissues and organs. CRISPR/Cas has tremendously advanced our understanding of hereditary diseases by enabling the rapid generation of novel cellular and animal models. Furthermore, CRISPR/Cas has become a valuable and effective option for the treatment of many diseases and disorders which would otherwise be incurable. In this review, we aim to highlight the ongoing development of innovative gene-editing tools used with LV platforms in a broad range of research and clinical applications.

In nature, CRISPR/Cas systems act as a prokaryotic adaptive immunity mechanism to recognize, target, and destroy the foreign DNA and RNA of phages and viruses [43,44]. The CRISPR/Cas systems are very diverse, composing so far six Cas enzyme types (I–VI), with at least 30 subtypes [45,46]. Despite this diversity, the Cas family’s members share similar components, including a CRISPR-RNA, or guide RNA (gRNA), and in most cases a trans-activating RNA (tracrRNA), [47,48]. The most studied and developed system of CRISPR/Cas is derived from the class II CRISPR-associated enzyme Cas9, which operates as a single effector; in contrast, class I Cas enzymes act as a multi-subunit protein system (reviewed in [49]). In this review, we will focus only on CRISPR/Cas9 systems; for comprehensive reviews on other Cas proteins, we would refer the reader to the following publications [50]. To achieve genome-editing several important steps have to occur, leading to the assembly of both RNAs (gRNA and tracrRNA) into one CRISPR/Cas complex. The combined version of both, namely synthetic guide RNA (sgRNA), greatly simplifies the delivery and expression of CRISPR/Cas systems. Cas9 is capable of binding DNA only in the presence of a specific sequence, known as a protospacer-adjacent motif (PAM). The recognition of the PAM motif enables Watson–Crick RNA-DNA base pairing. Following assembly of the ternary complex between RNA, DNA, and Cas9, the endonuclease domains become active, and cleave within both DNA strands, which results in the formation of double-strand DNA breaks (DSBs). The unprecedented efficiency, accuracy, and specificity of the Cas9 protein has been rapidly recognized and directed towards a wide range of genome-editing applications, from basic science to translational and clinical research and medicine [51].

## 7. The Use of Active Cas9 for Genome-Editing Applications

As mentioned above, cleavage mediated by Cas9 results in the generation of DSBs (Figure 4). Eukaryotic organisms mainly repair DSBs through an error-prone non-homologous end joining (NHEJ) mechanism, which leads to the formation of small insertions or deletions (InDels) (Figure 4). Alternatively, if a repair donor template is provided, host-mediated repair machinery can activate the homology-directed repair (HDR) pathway, resulting in error-free replacement of the target sequence (Figure 4). Unfortunately, HDR is typically characterized by very low efficiency [52]. In fact, it is widely accepted that the rate of HDR events range between 0.1 and 1%, depending on the system and the target gene sequence [52]. The rate could be even lower in vivo [53]. Another drawback of the HDR pathway is that it is not active in non-dividing and terminally differentiated cells, including brain neurons. Furthermore, the DSBs needed to trigger more efficient HDR also increase the possibility of non-specific cleavages and recombinations, and even on-target HDR can have highly negative effects on the cells [54,55]. These limitations established a significant need for the development of single-base-pair editing and prime-editing technologies to enable precise genome editing in non-dividing cells and tissues (discussed in detail below and reviewed in [56,57]).

## 8. Deactivated Cas9 (dCas9) for CRISPRi and CRISPRa Approaches

The ability of Cas9 to bind to specific sequences with strong affinity is of immense value in and of itself, independently of its endonuclease-enzymatic activity. Cas9 is a modular enzyme that possesses well-characterized catalytic domains. The amino acid residues that are necessary for the endonucleic activity of the enzyme are located in the RuvC and HNH domains [48,58]. It has been demonstrated that D10A and H840A substitutions in the RuvC and HMH domains, respectively, greatly reduce the catalytic activity of the enzyme without compromising the ability of the deactivated enzyme to bind to the targeted DNA sequence [48,58]. This mutated version of Cas9 is called deactivated or dead; implying that it supports no endonucleatic activity. Deactivated Cas9 has been widely used for transcriptional and epigenomic targeting. For example, dead Cas9 (dCas9) linked to a transcription repressor (e.g., KRAB) can be used for targeted repression of a gene-of-interest. Similarly, gene activation can be achieved using a dCas9- activator fusion variant (reviewed in [59]). The CRISPR/dCas9 systems used for gene activation and repression have been coined CRISPR-activation (CRISPRa) and CRISPR-interference (CRISPRi), (Figure 5). The basic repression system includes a dCas9-KRAB fusion; conversely, activation can be achieved using dCas9 fused to the VP64 effector (Figure 5), (reviewed in [60]). These basic tools paved the way for the development of more robust and efficient activation and repression systems, including SAM and SunTag [61]. The SunTag system, developed by Tanenbaum and colleagues in [61], utilizes the interaction between 10–24 copies of the short epitope GCN4 and its cognate scFV antibody expressed from a separate plasmid [61]. Placing dCas9-GCN4 on one vector and various effectors, activators or repressors fused with the scFV antibody on the other vector, enabled the high-affinity interaction, resulting in functional activation or repression. The reported levels of the activation and repression are significantly higher than those achieved with more basic systems [61]. Alternative to the SunTag system, the human CRISPR/Cas9 synergistic activation mediator (SAM) approach has been developed in Feng Zhang’s laboratory [62]. Similar to SunTag system, SAM uses several strong transcriptional activators linked together to amplify the effect. These activators are VP64, heat-shock factor 1 (HSF-1), and the p65 subunit of NF-kappaB (NF-kB) [63]. It is important to note that CRISPRi- and CRISPRa-mediated outcomes are reversible in general, as both systems target chromatin structure rather than the DNA sequence. Obviously this feature is appealing for many gene therapy applications. We recently developed a novel epigenome-editing platform based on an all-in-one LV for targeted DNA methylation editing within intron 1 of the alpha-synuclein-encoding gene, *SNCA*. Dysregulation of SNCA expression is one of the causative factors for Parkinson’s disease (PD) [64]. The system consists of dCas9 fused with the catalytic domain of DNA-methyltransferase 3A (DNMT3A). Applying the system to human-induced pluripotent stem cell (hiPSC)-derived dopaminergic neurons from a Parkinson’s disease patient with the *SNCA* triplication resulted in robust and steady downregulation of *SNCA* mRNA and protein levels, mediated by targeted DNA methylation at a regulatory sequence located in intron 1 of *SNCA*. We further demonstrated that this reduction in *SNCA* levels is capable of rescuing disease-related phenotypes. This developed approach suggests broad potential for this target sequence combined with LV-CRISPR-dCas9 technology as a novel epigenetic-based therapeutic strategy for various cases of PD [65].

## 9. Base-Pair Editing Technology

Generally, the most common genetic variants associated with hereditary diseases in humans are point mutations and functional single-nucleotide polymorphisms (SNPs) characterized by high-level penetration [66]. Therefore, gene-editing systems capable of inducing a robust, efficient, and safe conversion on a single nucleotide level have the potential to cure many hereditary diseases. The construction of a cytosine base-editor (CBE) system by David Liu’s group was the first important development towards the establishment of such tools (Figure 6A). In the paper published by Komor and colleagues, dCas9 was fused with rat APOBEC1, a cytosine deaminase enzyme [67]. The resulting system induces conversion of all cytosines to uracils within an approximately six-nucleotide window, counted from the PAM-20 to the left (upstream). The uracil is then read as thymine during replication, completing the C-to-T conversion [67]. To overcome a caveat associated with the activation of base excision repair, catalyzing U-to-C re-conversion, the second generation of CBE carried a uracil glycosylase inhibitor domain to deactivate the base-excision repair pathway. This improvement greatly enhances the efficiency of the base editor. The use of nickase Cas9 (nCas9) to add cuts in non-edited strands further improved CBE efficiency. The new system has been coined BE3 and has been proven to be highly efficient in a variety of human cells, with a correction rate ranging between 25–75%. It is important to note that the nickase approach also results in an unwanted increase in the formation of InDels, from less than 0.1% to approximately 1% [67]. More recently, base-editing systems have been further enhanced. For example, Komor et al. created the BE4-Gam system, carrying a second copy of the uracil glycosylase inhibitor and a bacteriophage protein, Gam. Gam has a high affinity for the free ends of DSBs, thus preventing NHEJ-mediated repair and reducing InDel formation [68]. Koblan et al. inserted two nuclear localization signals (NLS), placing them proximal and distal to nCas9. In addition, they evolved a more efficient deaminase by constructing a codon-optimized version of the APOBEC domain, yielding BE4max and ancBE4max [69]. Other groups have focused on limiting or expanding the cytosine deaminase activity window, allowing for C-to-T conversions within a window as short as three or as long as twelve nucleotides [70].

The development of an adenosine base editor (ABE) platform supporting A-to-G transitions vastly broadened the applications of base editing (Figure 6B). The original ABE platform was created by Gaudelli and colleagues, who connected nCas9 with an evolved deoxyadenosine deaminase, which catalyzes an adenosine to inosine transition [71]. Like the two-step CBE reaction, the inosine is then read as guanine during replication, completing the A-to-G transition. The efficiency of the ABE7 system engineered by Gaudelli et al. [71] was 50% in HEK293 cells, with an InDel formation rate of less than 0.1%. Notwithstanding this success, ABE7 was found to be incompatible with Cas9 of any species other than *Streptococcus pyogenes* (*Sp*Cas9). The incompatibility is due to the low DNA-bound residence-time of non-*Sp*Cas9, coupled with the decelerated enzymatic pace of deoxyadenosine deaminase [72]. To circumvent this bottleneck, Richter and colleagues used phage-assisted-continuous evolution (PACE) and phage-assisted non-continuous evolution (PANCE) methods to evolve the ABE8e system, which shows a greatly accelerated catalytic rate for the deoxyadenosine deaminase process, resulting in substantially higher efficiency [72]. ABE8e also displays increased processivity, which could be particularly advantageous for the multiplexed perturbations. Further improvement of the ABE system by Gaudelli and colleagues created an array of new eighth-generation ABEs, characterized by increased activity and editing efficiency and a broader window of editing [73]. This novel system has shown success in an adult mouse model of Duchenne muscular dystrophy, by enabling correction of the *DMB* gene in 17% of myofibers, with no measurable InDels or other off-target effects. It has to be pointed out that this rate of gene correction is therapeutic, considering the threshold level of 4% of the gene expression that is needed to correct muscle function [73]. Most recently, three different groups reported the development of CRISPR-Cas9-based dual programmable adenine and cytosine editors, capable of introducing A-to-G and C-to-T substitutions at the same time, with minimal off-target edits [74]. These dual CBE/ABE systems expand the range of possible DNA sequence alterations, broadening the research applications of CRISPR base editors.

## 10. Prime-Editing Technology

The majority of mutations linked to human genetic diseases are transitions; nevertheless, according with ClinVar database, correcting about 40% of the hereditary diseases would require transforming a purine-base nucleotide into pyrimidine-base nucleotide or vice versa. Thus, the bottleneck of the base-editing technology was recently addressed by David Liu’s group, who found a very elegant way to address this problem. In their recent publication, entitled “Search-and-replace genome editing without double-strand breaks or donor DNA,” Anzalone et al. described a new method of genome editing based on the rewriting genetic information into a specified DNA site using a dCas9 or nCas9 fused to an engineered viral reverse transcriptase (RT), paired with a prime editing guide RNA (pegRNA) that both specifies the target sequence and encodes the edit-of-interest [57] and (Figure 7). The authors validated these “prime editing” tools in human cells by correcting mutations in sickle cell disease and Tay-Sachs disease (efficiently and with few byproducts), generating a protective transversion in PRNP, and introducing various epitopes and tags into targets-of-interest [57]. Prime editing has demonstrated comparable or higher efficiency than the HDR-based equivalent and has had complementary strengths and weaknesses compared to base editing [57]. Consistent with these observations, we demonstrated that prime-editing system delivered by lentiviral vector is capable to correct any mutations precisely and specifically, albeit with low efficiency (personal communications). Nevertheless, in our hands, the efficacy of the prime editing was found to be significantly higher than that of the HDR. In fact, we found that the rate of the mutations introduced using the prime-editing approach ranges between 5 and 15% depending on the used system (personal communications). We believe that the low editing capacity of the prime-editing technology could stem from the fact that the expression level of Cas9-RT is fairly reduced, similar to the expression of base-editing tools discussed above (personal communications). As such, further evolution of the prime editing system will be necessary to address its shortcomings. Overall, prime editing substantially expands the scope and capabilities of genome editing, and in principle could correct up to 89% of known genetic variants associated with human diseases [57]. A good summary expanding on the diversity of dCas9-effector tools and systems could be also found in Rittiner et al. [41].

## 11. Lentiviral Vectors Paired with Genome-Editing Tools

Effective gene-editing using the CRISPR-Cas9 system can only be achieved through efficient delivery of all elements into target cells. Given the proven efficacy and the improved safety of lentiviral vectors for the delivery of therapeutic genes, lentiviral delivery platforms were among the first to be developed and adapted for genome-editing applications In fact, as soon as CRISPR-Cas9 system was reported to function in human cells, all-in-one lentiviral vectors expressing both Cas9 and sgRNAs were designed and constructed [29]. In principle, CRISPR/Cas9 vector systems follow the same organization as those delivering short hairpin RNA (shRNA). The expression of both systems is usually driven by pol III promoter, e.g., U6 and H1, and terminated by pol III poly-adenylation signal. In contrast, the Cas9 enzyme could be expressed from a strong ubiquitous pol II promoter, e.g., CMV and EF1-α, or an inducible counterpart, e.g., tetracycline-regulated promoter, Ptet, which comprised several repeats of tet-operator (TetO) sequences placed upstream of a minimal promoter such as the CMV promoter [75]. Similar to the shRNA technology, lentiviruses paired with CRISPR/Cas9 systems originally have been developed in the format of genomic libraries consisting of thousands of sgRNAs [76]. Such CRISPR libraries have been used in a genome-wide loss of function screen [29]. In fact, Shalem et al. demonstrated that lentiviral delivery of a genome-scale CRISPR-Cas9 knockout (GeCKO) library targeting 18,080 genes with 64,751 unique sgRNA sequences enables both negative and positive selection screening in human cells. The authors used the GeCKO library to identify genes essential for cell viability in cancer and pluripotent stem cells, as well to screen for genes whose loss is involved in resistance to vemurafenib, a therapeutic RAF inhibitor using melanoma cells as a model [29]. Furthermore, lentiviral vectors paired with CRISPR/Cas9 library were designed to support a functional screening identified novel tumor suppressor genes involved in acute leukemia [77]. On a smaller, lentivirus-based array- screen scale, overlapping sgRNAs have been produced and pooled to enable a mutational analysis of key sequence cis-acting elements directing the transition from fetal to adult hemoglobin. Importantly, in this study, Canver et al., validated BCL11A erythroid enhancer as being a target for fetal hemoglobin reinduction [78]. Importantly, lentiviral vectors paired with CRISPR/Cas9 were developed and used towards treatment of many infectious diseases, including HIV-1, HBV, and HSV-1, as well as to correct defects in underlying human hereditary diseases, including cystic fibrosis, and many neurodegenerative diseases and disorders [79]. In addition to the ability of LVs to efficiently deliver active Cas9-CRISPR-systems, LVs have been successfully paired with dCas9-DNA deaminases (the base-editing approach), and epigenome-modified enzymes including histone deacetylases and other chromatin modifiers [80]. As pointed out earlier in the discussion, long-lasting expression of lentivirus-CRISPR/Cas9 tools may increase the risks of undesirable off-target effects, characterized by non-specific RNA–DNA interactions and off-target DNA cleavages or other off-target perturbations, depending on the expressed Cas9-effector. The IDLV vectors described above present an attractive opportunity to improve the delivery system by enabling transient expression of CRISPR/Cas components. To take advantage of the transient nature of the IDLV viral platform, we recently developed an all-in-one IDLV vector carrying the improved and optimized expression cassette. We demonstrated that the new delivery platform could generate significantly higher titers and support the greater expression of CRISPR/Cas transgenes, compared to the previously used ICLV and IDLV vectors. In addition, we reported that the novel vector was capable of mediating a robust and efficient level of CRISPR/Cas9 dependent gene-editing in post-mitotic neurons, both in vitro and in vivo [37]. We then used the same backbone to modulate pathogenic overexpression of the SNCA gene, elevated levels of which are implicated in Parkinson’s disease (PD) [64]. To that end, we focused on DNA methylation within SNCA intron 1, which regulates SNCA transcription. To target hypomethylated intron 1 DNA sites, we developed a system comprising an all-in-one lentivirus paired with CRISPR-dCas9, which was fused with the catalytic domain of DNA-methyltransferase 3A (DNMT3A) [64]. Applying the system to human-induced pluripotent stem cell (hiPSC)-derived dopaminergic neurons from a PD patient with the SNCA triplication, we found a significant downregulation of SNCA mRNA and protein as a result of targeted DNA methylation at intron 1. Importantly, the all-in-one system targeting SNCA intron 1 was capable of rescuing disease-related cellular phenotypes characteristic of SNCA triplication in hiPSC-derived dopaminergic neurons, including mitochondrial ROS production and cellular viability, among others [64]. This work established DNA methylation at SNCA intron 1 to be a therapeutic target which could be used to combat overexpression of the gene involved in Parkinson’s disease pathology. Along the same lines, we most recently demonstrated a rescue of the PD phenotype in isogenic hiPSC cells derived from PD patients with an autosomal dominant SNCA mutation A53T. We demonstrated that cells targeted with LV-CRISPR/Cas9-DNMT3A showed substantial improvement in PD phenotypes, including nuclear transport, nuclear shape and integrity of the nuclear envelope, resistance to various stress stimuli, and others [65]. The optimized protocol for production of a lentivirus-CRISPR/dCas9 system at high-titers for transduction into primary cells, including hiPSCs and NPCs used in the above experiments, was published in [81]. The protocol describes how to produce, purify, and concentrate lentiviral vectors and to highlight their suitability for epigenome- and genome-editing applications.

## 12. Conclusions

CRISPR/Cas technology has revolutionized all aspects of modern life science research, ranging from basic science to clinical research. Importantly, recent progress in lentiviral vectorology has fully supported and enabled robust and efficient delivery of CRISPR/Cas tools into tissues- and organs-of-interest in vivo, opening doors for novel approaches and strategies previously unimaginable. While these innovative technologies and tools continue to progress, several key issues still need to be solved. One of the issues is the relatively high levels of undesirable off-target effects induced as the consequence of permanent expression of CRISPR/Cas9 tools delivered by LVs. In addition to causing undesirable effects outside of the targeted gene or region, integrating lentivirus (mostly meaning simple retroviruses, e.g., γ-retrovirus) is capable of inducing measurable levels of toxicity and pathogenicity, stemming mostly from its oncogenic potential. Unfortunately, other viral delivery platforms are not fully detached from the adverse effects generally associated with viruses and may cause tragic outcomes. [71,72,73]. Adverse reactions resulting from the integrating nature of WT-LVs could be minimized with transient, non-integrating vectors, e.g., IDLVs. Indeed, we recently reported that IDLVs demonstrate significantly lower levels of InDel formation and other off-target effects compared to their integrase-competent counterparts, both in vitro and in vivo [37,39]. Another potential issue is that permanently expressed, host genome-integrated lentivirus may be a target for mobilization by incoming non-infectious (vector) or infectious (HIV-1) viral particles in the case of superinfection. Vector mobilization may lead to even higher levels of integration events, and eventually to the insertional mutagenesis, oncogenic activation, and other off-target perturbations mentioned above. From this perspective, IDLVs would also be preferred over WT-LVs, as IDLVs have demonstrated lower levels of vector-with-vector and vector-with-virus mobilization events [21]. However, further improvement in lentivirus design aimed to create a recombinogenic-less vector will be needed to fully address this concern [17]. Lastly, as disease progression is often dynamic over time, it will be important to improve spatial- and time-regulation of target gene expression to achieve more accurate and fine-tuned treatment.

## Figures and Tables

**Figure 1 viruses-13-01288-f001:**
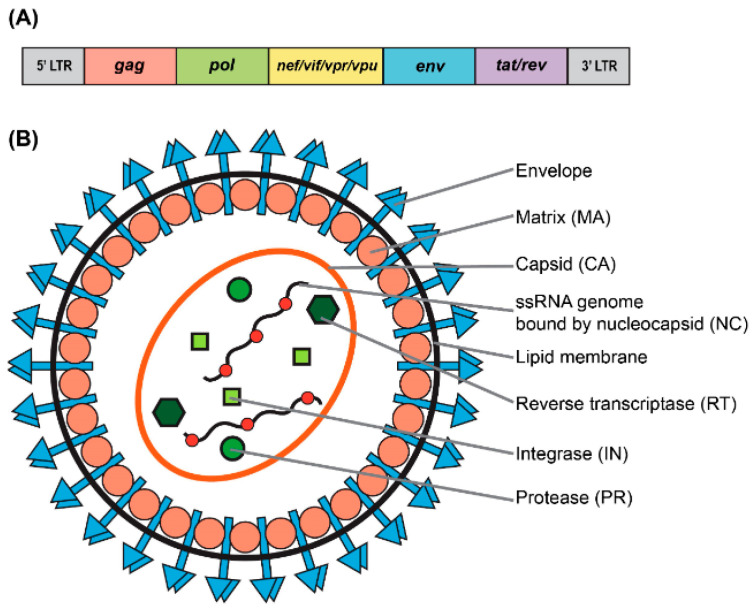
Genomic and structural organization of lentiviral vector. (**A**) Simplified schematic of the wild-type human immunodeficiency virus type-1 (HIV-1) genome. (**B**) Particle structure of a lentivirus.

**Figure 2 viruses-13-01288-f002:**
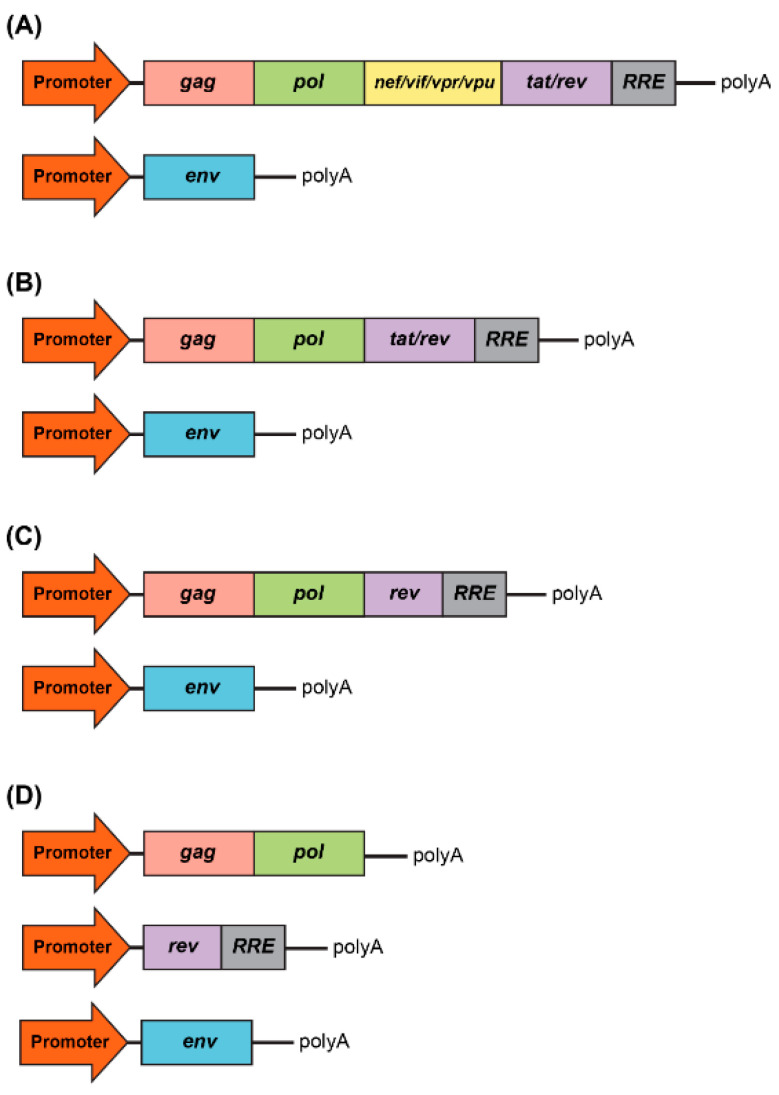
Development of packaging system for lentiviral production. (**A**) First generation included all accessory genes, nef, vif, vpr, and vpu, and the regulatory proteins, tat and rev. RRE is the rev response element. (**B**) Second generation excluded all accessory genes, nef, vif, vpr, and vpu. (**C**) Third generation excluded the tat regulatory protein. (**D**) Fours generation is characterized by the split of the gag/pol and rev sequences into two different cassettes, which evidently benefits the vector safety.

**Figure 3 viruses-13-01288-f003:**
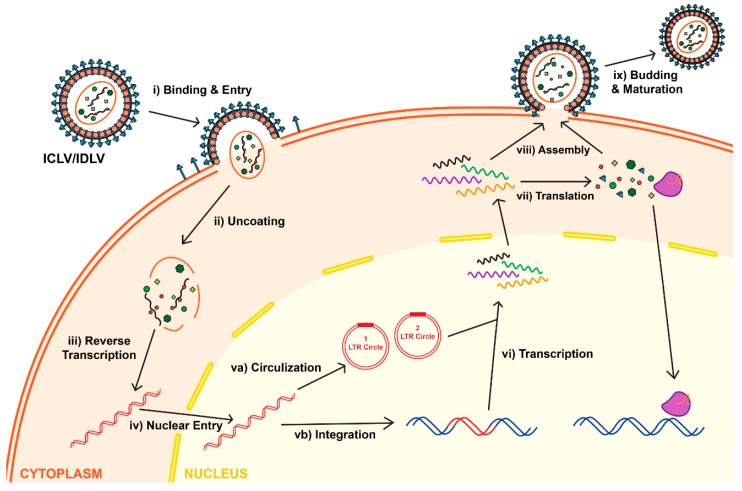
Life cycle of lentiviruses for both Integrase-Competent Lentivirus (ICLV) and Integrase-Deficient Lentivirus (IDLV). Initially, IDLVs and ICLVs both bind and enter target cells (**i**). Uncoating of lentiviruses exposes the viral RNA (**ii**), allowing reverse transcription to occur in the cytoplasm (**iii**). The dsDNA product is then imported into the nucleus (**iv**). Some of this dsDNA integrates into the host genome (**vb**), while the majority recombines into one- or two-LTR circles and remain episomal (**va**). Due to the differences of the Integrase protein, the rates of integration and circle formation differ between IDLVs and ICLVs. The viral dsDNA undergoes transcription (**vi**), which is then translated in the cytoplasm into lentiviral structural proteins and gene of interests, such as the CRISPR/Cas system (**vii**). The viral RNA and structural protein assemble (**viii**), which then undergoes budding and maturation to form a new lentivirus (**ix**).

**Figure 4 viruses-13-01288-f004:**
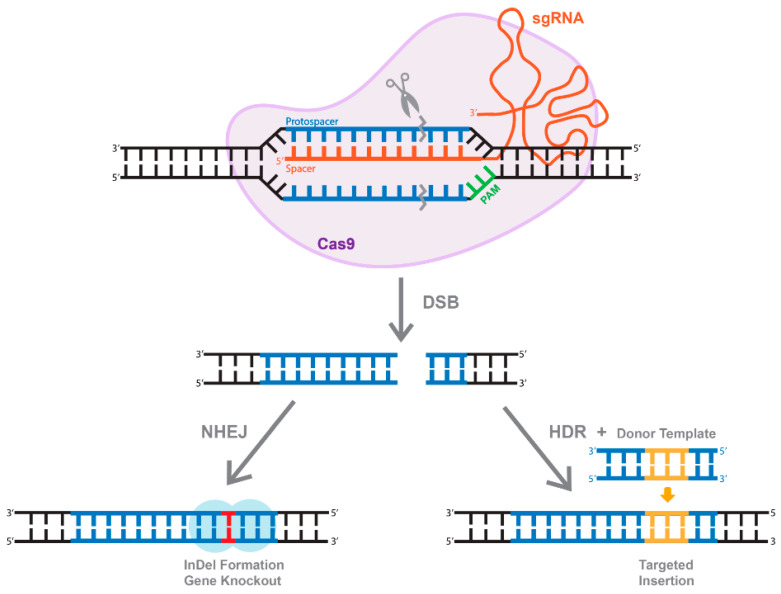
The overview of CRISPR/Cas system. Active Cas9 enzyme introduces a double-stranded break (DBS). The cell repairs through two methods, one of which is through non-homologous end joining (NHEJ) that create InDels. Alternatively, of a dsDNA donor template is provided, the DSB is repaired through homologous recombination (HDR), resulting in a targeted insertion.

**Figure 5 viruses-13-01288-f005:**
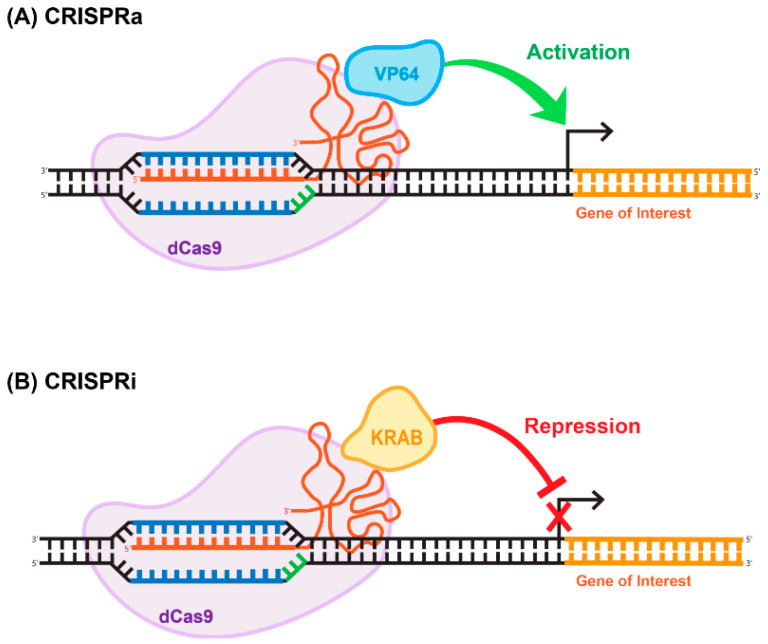
Repurposing CRISPR/Cas for epigenome-editing applications. (**A**) CRISPR-activation (CRISPRa) often consists of a dCas9 fused directly to a single transcriptional activator (e.g., VP64), which then activates the gene of interest. (**B**) Similarly, CRISPR-interference (CRISPRi) often consists of a dCas9 fused directly to a single transcriptional repressor (e.g., KRAB), which then represses the gene of interest.

**Figure 6 viruses-13-01288-f006:**
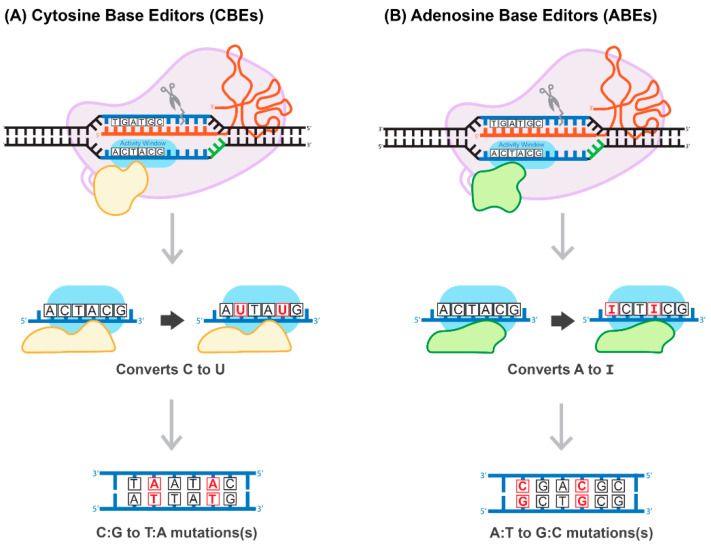
Proposed mechanism of base-pair editing CRISPR/Cas technology. (**A**) Cytosine Base Editors (CBEs) catalyze the conversion of all cytosines to uracils within a 5-6 activity window. Uracil is read as thymine during replication, which then converts all C:G to T:A. (**B**) Adenosine Base Editors (ABEs) catalyze the conversion of all adenosines to inosines within a 5-6 activity window. Inosine is read as guanine during replication, which then converts all A:T to G:C.

**Figure 7 viruses-13-01288-f007:**
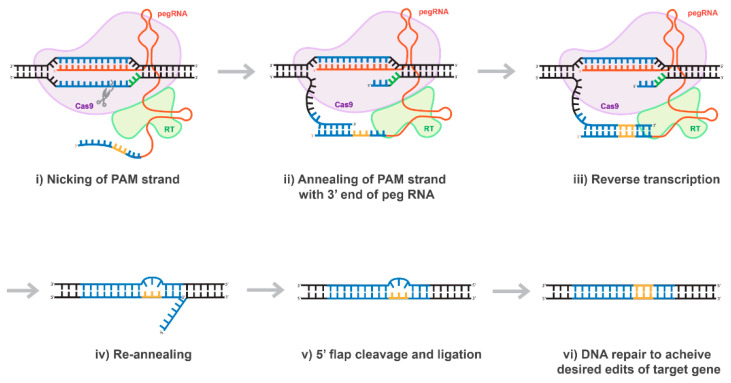
Proposed mechanism of prime editing CRISPR/Cas technology. Once the 5’ end of the pegRNA (spacer) binds to the protospacer of the target DNA, the protospacer-adjacent motif (PAM) strand of the target DNA is nicked (**i**). The nicked PAM strand hybridizes with the primer binding site (PBS) on the 3’ end of the pegRNA (**ii**). The pegRNA serves as a template as the reverse transcriptase (RT) fused to the Cas9 enzyme extends the 3’ end of the nicked PAM strand (**iii**). The prime editing CRISPR/Cas system disengages and the target site is left with a 3’ flap (the edited PAM strand) and a 5’ flap (the original PAM strand) (**iv**). The 5’ flap is preferentially degraded by cellular endonucleases and the edited PAM strand ligases and hybridizes with the non-PAM strand (**v**). Finally, through DNA repair mechanisms, the desired edits of the target gene are transferred to the non-PAM strand (**vi**).

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
