# Peer review of "Lentiviral Vectors for Delivery of Gene-Editing Systems Based on CRISPR/Cas: Current State and Perspectives"

_viruses, 2021, doi:10.3390/v13071288_

Round 1
Reviewer 1 Report
The review by Dong and Kantor presents a concise review of advances in CRISPR/Cas systems and also talks about how non-integrating lentiviral vectors may be used to assist in the delivery of CRISPR/Cas. In general, the review is well written and the figures are generally clear. This will be of interest to investigators in this area. There are a number of statements which appear to be unsupported or overstatements. While the review is focused on lentiviral vectors, it should at least have a section which discusses alternative systems that can deliver CRISPR/Cas. The authors should consider rewording their title as it is misleading. There is an excellent discussion of novel CRISPR/Cas systems which is the bulk of the paper. There is less than a page of lentiviral vectors paired with genome editing tools.
Major Concerns:
1) Line 67. I believe “common use” is an overstatement. I agree that some investigators are using second generation constructs but they are in the minority due to potential biosafety concerns. Second generation vectors are also not favored by regulators. IBCs can also require more rigorous screening for replication competent viruses. It is insufficient to make this statement with only a personal communication, if they authors would like to continue to make this statement clear references that can provide supportive data should be included.
2) The authors provide good information on packaging systems but little information about lentiviral vector design. This should be included, and description of vector design, including SIN-LTR and the safety implications (compared to gamma retroviral vectors).
3) Line 125. In general, the discussion of insertional mutagenesis does not give a fair representation of the safety differences between gamma retroviral vector and HIV-based lentiviral vectors. The use of MLV enhancers in gamma retroviral vector present significant risks in hematopoietic stem and progenitors but this has not proven to be the same risk for lentiviral vectors. Agreed that there is a risk with lentiviral vectors, but the risk is significantly different from gamma retroviral vectors both in in vitro models and in clinical trials.
4) Line 126. The authors state that the risk of insertional mutagenesis has hampered the use of lentiviral vectors in clinical research. The references are from 2014 but since then there has been an ever increasing number of clinical trials using lentiviral vectors for cancer (CAR-T) and genetic diseases. Since the authors are promoting IDLV, a reader may not grasp that integrating vectors are used to permanently alter a target cell, and IDLV+CRISPR/Cas aims to do this with less risk of mutagenesis. Since the reader may be new to the field, a better explanation of these issues is warranted. It is also relevant that the US FDA has approved products based on gamma retro and lentiviral vectors.
5) Line 135. The relevance of a live EIAV in risk of HIV based non-replication competent system is unclear. The authors need to give a fuller explanation of replication competent and incompetent virus risks for a reader of the review to understand the point being made.
6) Line 382. I believe there are many investigators, including those in the AAV and nanoparticle field, that would take issue to the statement that lentiviral is the “platform of choice” for delivering CRISPR/Cas. The lack of discussion of alternative methods for delivering editing systems is a major deficiency in this review. While the extensive details of the other systems is not expected for this review, a discussion of the pluses and minuses would provide for a balanced review.
Minor Concerns:
1) Line 92. Fourth not Fours
2) Similar to the ….
3) Line 357 – Consistent not Consistently
Author Response
Reviewer 1
The review by Dong and Kantor presents a concise review of advances in CRISPR/Cas systems and also talks about how non-integrating lentiviral vectors may be used to assist in the delivery of CRISPR/Cas. In general, the review is well written and the figures are generally clear. This will be of interest to investigators in this area. There are a number of statements which appear to be unsupported or overstatements. While the review is focused on lentiviral vectors, it should at least have a section which discusses alternative systems that can deliver CRISPR/Cas. The authors should consider rewording their title as it is misleading. There is an excellent discussion of novel CRISPR/Cas systems which is the bulk of the paper. There is less than a page of lentiviral vectors paired with genome editing tools.
We thank the reviewer for these comments and suggestions. The title of the manuscript has been changed to the following: “Lentiviral vectors for delivery of gene-editing systems based on CRISPR/Cas: current state and perspectives”. We included a section to discuss alternative system used for the delivery of CRISPR/Cas, specifically, adeno-associated vectors (AAV). In addition, we extended discussion of lentiviral vectors paired with genome editing tools.
Major Concerns:
- Line 67. I believe “common use” is an overstatement. I agree that some investigators are using second generation constructs but they are in the minority due to potential biosafety concerns. Second generation vectors are also not favored by regulators. IBCs can also require more rigorous screening for replication competent viruses. It is insufficient to make this statement with only a personal communication, if they authors would like to continue to make this statement clear references that can provide supportive data should be included.
Our reply: We thank the reviewer for this comment and agree with this suggestion. We deleted the sentence, “ it has to be noted that the second-generation of the packaging system is still in the common use, as carrying the tat gene seems to be advantageous for the viral production and titers (personal communications)” from the text.
- The authors provide good information on packaging systems but little information about lentiviral vector design. This should be included, and description of vector design, including SIN-LTR and the safety implications (compared to gamma retroviral vectors).
Our reply: We thank the reviewer for this comment. We added the below paragraph entitled “Safety of lentiviral vectors” to extend on the discussion of the LV design.
Safety of lentiviral vectors As mentioned above, the use of heterologous promoters, e.g. CMV or RSV instead of the endogenous HIV-1 promoter located in the 5′-LTR, enabled an independence from the Tat protein, which greatly improved the vector’s safety [17], [18, 19], [5], [9]. Subsequently and importantly, deletions within the U3′-region of the 3-LTR that harbors the enhancer/regulatory elements′ and the TATA box, helped to develop a self-inactivating (SIN) lentiviral vector [20], [18], [21]. During RT reaction, the U3’ region relocates to the 5′-LTR, as such SIN-vectors are completely devoid of HIV-1 enhancer/ promoter sequences. Consequently, SIN-vectors lack the ability of producing a full-length vRNA that could be packaged into the viral particles. It would be important to note that the deletion did not affect vector production, and vector titer remains to be comparable with those of non-SIN counterparts. As mentioned above, development of a SIN vectors further reduced the likelihood of the appearance of RCVs. Furthermore, it greatly lower a likelihood of mobilizing the vector’s mRNA by the replication-wt virus [22], [23]. In addition, lack of enhancer/promoter region diminished the risk of inadvertent activation of silent host promoters by the lentiviral provirus.
- Line 125. In general, the discussion of insertional mutagenesis does not give a fair representation of the safety differences between gamma retroviral vector and HIV-based lentiviral vectors. The use of MLV enhancers in gamma retroviral vector present significant risks in hematopoietic stem and progenitors but this has not proven to be the same risk for lentiviral vectors. Agreed that there is a risk with lentiviral vectors, but the risk is significantly different from gamma retroviral vectors both in in vitro models and in clinical trials.
Our reply: We agree that the discussion related to the insertional mutagenesis does not give a fair representation of the safety differences between gamma retroviral vector and HIV-based lentiviral vectors. To address this, we highlighted that the issue is related to the simple retroviruses, highlighting the following: “Despite significant improvement in viral safety, simple retroviruses, e.g. γ-retroviruses have a relatively high risk of insertional mutagenesis (reviewed in [2], [24]). In fact, initially successful treatments of ADA-SCID, SCID-X1, and X-linked CGD diseases with γ-retroviral vectors were tragically complicated by blood cancers developed by several patients [25], [26], [27]. It has been reported that the patients carried provirus DNA in the proximity of proto-oncogenes dysregulating their expression” [25], [26], [27].
To further stress the difference, we added the following sentence, “Generally, LV has the same traits, due to the inherent capacity for integration; with that said, the risk of oncogenicity and toxicity of the complex retroviruses seems to be lower than of that of simple retroviruses, e.g. γ-retroviruses”.
- Line 126. The authors state that the risk of insertional mutagenesis has hampered the use of lentiviral vectors in clinical research. The references are from 2014 but since then there has been an ever increasing number of clinical trials using lentiviral vectors for cancer (CAR-T) and genetic diseases. Since the authors are promoting IDLV, a reader may not grasp that integrating vectors are used to permanently alter a target cell, and IDLV+CRISPR/Cas aims to do this with less risk of mutagenesis. Since the reader may be new to the field, a better explanation of these issues is warranted. It is also relevant that the US FDA has approved products based on gamma retro and lentiviral vectors.
Our reply: We agree with this comment and introduced the following paragraph to better address the concern.
“Importantly, both γ-retroviral vectors and lentiviral vectors have been recently developed into efficient platform utilized for curing monogenic inherited disorders caused by an altered development and/or function of the blood system, such as immune deficiencies and red blood cell and platelet disorders (reviewed in [31]). Thanks to the positive efficacy and safety data from hematopoietic stem progenitor cells (HSPCs) gene therapy trials collected during last decade, two advanced therapies based on γ-retroviral vector-engineered HSPCs have been approved for the EU market, and many other clinical trials are also in advanced stages in the US [31]). In 2016, the European Medicines Agency (EMA) approved Strimvelis for the treatment of ADA-SCID. a replication deficient γ-retroviral vector based on Moloney murine leukemia virus (MMLV) encoding the cDNA sequence for human ADA [32], [33]. Next, in 2019, Zynteglo became the first lentiviral vector- based gene therapy product for transfusion-dependent β-thalassemia patients approved by the EMA [34]. Furthermore, lentiviral-HSPC-based medicine, Libmeldy, most recently received EMA positive opinion for marketing authorization to treat metachromatic leukodystrophy (MLD) [31]”.
To make better transition to the IDLVs’ systems, we also introduced a comment related to the difference between stable transgene expression (desired in the above experiments) and transient transgene expression (sufficient and desired for gene-editing perturbations), as follows:
“The above applications required stable and long-lasting expression of the therapeutic transgenes supported by integrase-competent retro- and lentiviruses. Nevertheless, permanently expressed CRISPR/Cas systems being seen rather as substantial disadvantage, as stable expression of their components may facilitate undesirable off-target effects, hindering their utility for genome- and epigenome-editing applications [35]. Indeed, rise of promiscuous interactions with off-target genes due to excess guide RNA (gRNA)/Cas9 is well-documented [35], [36]. Furthermore, sustained expression of gRNA/Cas9 in vitro increases the tolerability of mismatches in the guide-matching region and the protospacer adjacent motif (PAM), thereby promoting non-specific double-strand breaks (DSBs). [37], [38]. Along the same lines, the ratio of insertions and deletions (indels) at off-target versus target sites in vivo increases with higher Cas9 and gRNA concentrations [39]. These results suggest that transient delivery systems utilizing a “hit and run” strategy for terminal modification of the DNA loci would be advantageous for high-precision gene editing”.
- The relevance of a live EIAV in risk of HIV based non-replication competent system is unclear. The authors need to give a fuller explanation of replication competent and incompetent virus risks for a reader of the review to understand the point being made
Our reply: We thank the reviewer for this comment and agree that the further clarification is needed to address the above confusion. As such, we introduced the following: “it has been demonstrated that the lentiviral vectors of horses, equine-infectious anemia virus (EIAV) vectors) can cause multiple tumors in the livers of mice following in utero and neonatal administrations [30]. A causal link between EIAV-vectors and tumorigenesis has yet to be established; however, it is important to note that in the same study the use of HIV-1 based vectors were not linked with formation of any detectable tumors [30]”.
- I believe there are many investigators, including those in the AAV and nanoparticle field, that would take issue to the statement that lentiviral is the “platform of choice” for delivering CRISPR/Cas. The lack of discussion of alternative methods for delivering editing systems is a major deficiency in this review. While the extensive details of the other systems is not expected for this review, a discussion of the pluses and minuses would provide for a balanced review.
Our reply: We agree with this comment and the reviewer’s suggestion to discuss on alternative methods for delivering gene-editing systems. To create more balanced review, per the reviewer’s recommendation, we included a section related to design, construction and use of AAV vectors for delivering gene-editing components. The introduced text is below:
Adeno-Associate Vectors (AAVs) This review is devoted to lentiviral vectors; however, it would be remiss not to mention the most frequently used viral platform for gene therapy, adeno-associated vector (AAV) (reviewed in [2], [24]). AAV is an ideal viral system for several reasons: (i) the vector has no known associated pathologies and causes only a mild immune response in humans; (ii) similarly to IDLV vectors, the AAV genome can be persisted long-termly in episomal forms, and thus presents an opportunity for extended transgene expression in non-dividing cells and tissues [2]. (iii) Lastly, the AAV genomic structure is well-characterized, so the consequences of genome-targeted manipulations can adequately be predicted [2]. [24] For the above considerations, over the last three decades a significant effort has been devoted to developing AAV into one of the gold-standard delivery systems for broad range of gene-therapy applications [24]. Notwithstanding these advances, impressive and rapidly diversifying array of AAV-CRISPR/Cas-derived tools predominantly have been used in vitro (reviewed in [47]). Efficient delivery in vivo using AAV vectors is a significantly more challenging task. In fact, the limited packaging capacity of the AAV genomes is the main bottleneck for its use, in an all-in-one configuration, for delivery of bulky and complex CRISPR/Cas transgenes in vivo. One of the approaches to overcome the significant restraints imposed by AAV’s ~4.7kb functional packaging capacity, is to physically split a CRISPR/Cas transgene into two pieces, which are packaged into separate AAV vectors. The resulting AAVs are then co-delivered, and the complete protein is reassembled in situ by a split intein – a pair of domains which “splice themselves out”, thus re-joining two peptides in the end-to-end configuration [48], [49], [50]. Nevertheless, further improvement of this and other systems (reviewed in [47]) would be necessary to achieve the desired therapeutic efficacy. As mentioned above, we also expanded on description of lentiviral vectors paired with genome editing tools.
Minor Concerns:
1) Line 92. Fourth not Fours
2) Similar to the ….
3) Line 357 – Consistent not Consistently
Our reply: Thank you for identifying these inconsistencies. We made the required corrections.
Reviewer 2 Report
This is a very comprehensive review summarizing lentiviral gene delivery systems and various applications of CRSPR/Cas systems. In particular, the authors discuss about new gene editing technologies in depth. This would be a good introduction for those who plan to employ these systems. With a minor revision, I recommend this manuscript to be published in Viruses.
Specific Points.
- I feel a bit of "disconnection" between the first part (lentiviral vector systems) and the second part (CRSIPR/Cas and its application). An additional paragraph/chapter describing lentiviral systems particularly designed for CRISPR/Cas experiments would be helpful. For example, how Cas9 and guide RNA are introduced/expressed to ensure all cells receive both? In one vector or separate lentiviral vectors with the same tropism? etc...
- Also, some examples of actual procedures for producing high titre viruses and infecting them to "difficult" cells (for example, neurons) would be highly informative
- Also, including descriptions about introducing guide-RNA libraries (instead of guide-RNA of targeted genes) would be informative.
- Since gene editing technologies using (d)Cas9-fusion proteins is a main topic of the review, including a table summarizing different fusions (fusion partners, activity, purpose/mechanism of action, and/or strength/weakness) would also be highly informative.
- Line 362: "between 5-to-15%" should be "between 5 and 15%".
- There are several "double-spaces" between words. Please go through the manuscript thoroughly and remove them. For example, lines 44, 74, 367, 396, 399 etc.
Author Response
Reviewer 2
This is a very comprehensive review summarizing lentiviral gene delivery systems and various applications of CRSPR/Cas systems. In particular, the authors discuss about new gene editing technologies in depth. This would be a good introduction for those who plan to employ these systems. With a minor revision, I recommend this manuscript to be published in Viruses.
- I feel a bit of "disconnection" between the first part (lentiviral vector systems) and the second part (CRSIPR/Cas and its application). An additional paragraph/chapter describing lentiviral systems particularly designed for CRISPR/Cas experiments would be helpful. For example, how Cas9 and guide RNA are introduced/expressed to ensure all cells receive both? In one vector or separate lentiviral vectors with the same tropism? etc...
Our reply: We agree with the comment that the manuscript could benefit from additional paragraph describing lentiviral systems particularly designed for CRISPR/Cas experiments. In this regard, we included the following paragraph introduced at the beginning of “Lentiviral vectors paired with genome-editing tools” section:
“Effective gene- editing using the CRISPR-Cas9 system can only be achieved through efficient delivery of all elements into target cells. Given the proven efficacy and the improved safety of lentiviral vectors for the delivery of therapeutic genes, lentiviral delivery platforms were among the first to be developed and adapted for genome-editing applications In fact, as soon as CRISPR-Cas9 system was reported to function in human cells, all-in-one lentiviral vectors expressing both Cas9 and sgRNAs were consrtructeddesigned and constructed [35], [91], [92]. In principle, CRISPR/Cas9 vector systems follow the same organization as those delivering short hairpin RNA (shRNA). The expression of both systems usually driven by pol III promoter, e.g. U6 and H1 and terminated by pol III poly-adenylation signal. In construct, Cas9 enzyme could be expressed from a strong ubiquitous pol II promoter, e.g. CMV and EF1-or inducible counterpart, e.g. tetracycline- regulated promoter, Ptet which comprised from several repeats of tet-operator (TetO) sequences placed upstream of a minimal promoter such as the CMV promoter [93]. Similar to the shRNA technology, lentiviruses paired with CRISPR/Cas9 systems originally have been developed in the format of genomic libraries consisting of thousands of sgRNAs [92]. Such CRISPR libraries have been used in a genome-wide loss of function screen [35], [91], [92]. In fact, Shalem et al. demonstrated that lentiviral delivery of a genome-scale CRISPR-Cas9 knockout (GeCKO) library targeting 18,080 genes with 64,751 unique sgRNA sequences enables both negative and positive selection screening in human cells. The authors used the GeCKO library to identify genes essential for cell viability in cancer and pluripotent stem cells, as well to screen for genes whose loss is involved in resistance to vemurafenib, a therapeutic RAF inhibitor using melanoma cells as a model [35]. Furthermore, lentiviral vectors paired with CRISPR/Cas9 library were designed to support a functional screening identified novel tumor suppressor genes involved in acute leukemia [94]. On a smaller, lentivirus-based array- screen scale overlapping sgRNAs have been produced and pooled to enable a mutational analysis of key sequence cis-acting elements directing the transition from fetal to adult hemoglobin. Improtantly, in this study Canver et al, validated BCL11A erythroid enhancer as being a target for fetal hemoglobin reinduction”.
In addition, we changed the transition sentence prior to redirecting the manuscript towards the description of the clinical work related to LV- CRISPR/Cas9 tools. The new sentence is as follows: “Importantly, lentiviral vectors paired with CRISPR/Cas9 were developed and used towards treatment of many infectious diseases, including HIV-1, HBV, and HSV-1, as well as to correct defects in underlying human hereditary diseases, including cystic fibrosis, and many neurodegenerative diseases and disorders [95], [96], [97], [98], [44], [77], [78], [60]”.
- Also, some examples of actual procedures for producing high titre viruses and infecting them to "difficult" cells (for example, neurons) would be highly informative
Our reply: We thank the reviewer for this comment and have added more information on the production of high-tittered lentiviruses to sustain an efficient transduction in challenging-for-transduction cells.
“The optimized protocol for production of lentivirus-CRISPR/dCas9 system at high-titers for transduction into primary cells, including hiPSCs and NPCs used in the above experiments published in [101]. The protocol describes how to produce, purify, and concentrate lentiviral vectors and to highlight their suitability for epigenome- and genome-editing applications”.
- Also, including descriptions about introducing guide-RNA libraries (instead of guide-RNA of targeted genes) would be informative.
Our reply: We thank the reviewer for this comment. We added the following paragraph to stress the significance of LV systems for introducing guide-RNA libraries: “Similar to the shRNA technology, lentiviruses paired with CRISPR/Cas9 systems originally have been developed in the format of genomic libraries consisting of thousands of sgRNAs [92]. Such CRISPR libraries have been used in a genome-wide loss of function screen [35], [91], [92]. In fact, Shalem et al. demonstrated that lentiviral delivery of a genome-scale CRISPR-Cas9 knockout (GeCKO) library targeting 18,080 genes with 64,751 unique sgRNA sequences enables both negative and positive selection screening in human cells. The authors used the GeCKO library to identify genes essential for cell viability in cancer and pluripotent stem cells, as well to screen for genes whose loss is involved in resistance to vemurafenib, a therapeutic RAF inhibitor using melanoma cells as a model [35]. Furthermore, lentiviral vectors paired with CRISPR/Cas9 library were designed to support a functional screening identified novel tumor suppressor genes involved in acute leukemia [94]. On a smaller, lentivirus-based array- screen scale overlapping sgRNAs have been produced and pooled to enable a mutational analysis of key sequence cis-acting elements directing the transition from fetal to adult hemoglobin. Improtantly, in this study Canver et al, validated BCL11A erythroid enhancer as being a target for fetal hemoglobin reinduction” [95]”.
- Since gene editing technologies using (d)Cas9-fusion proteins is a main topic of the review, including a table summarizing different fusions (fusion partners, activity, purpose/mechanism of action, and/or strength/weakness) would also be highly informative.
Our reply: We thank the reviewer for this comment. The requested topic can be found in our recently published review article entitled: “Gene-Editing Technologies Paired With Viral Vectors for Translational Research Into Neurodegenerative Diseases”. We referred the reader to this manuscript in the following sentence: “A good summary expanding on the diversity of dCas9-effector tools and systems could be also found in Rittiner et al. [60]”.
- Line 362: "between 5-to-15%" should be "between 5 and 15%". Corrected as requested
- There are several "double-spaces" between words. Please go through the manuscript thoroughly and remove them. For example, lines 44, 74, 367, 396, 399 etc.
Our reply: Thank you for identifying these inconsistencies. We went through the manuscript and made the required edits.
Round 2
Reviewer 1 Report
None